# Management of Female Genital Mutilation/Cutting-Related Obstetric Complications: A Training Evaluation

**DOI:** 10.3390/ijerph19159209

**Published:** 2022-07-28

**Authors:** Kim Nordmann, Ana Belén Subirón-Valera, Mandella King, Thomas Küpper, Guillermo Z. Martínez-Pérez

**Affiliations:** 1Institute of Occupational & Social Medicine, RWTH Aachen University, 52062 Aachen, Germany; tkuepper@ukaachen.de; 2Department of Physiatrics and Nursing, University of Zaragoza, 50009 Zaragoza, Spain; subiron@unizar.es; 3Research Group Water and Environmental Health (B43_20R), University Institute of Research in Environmental Science of Aragón, University of Zaragoza, 50009 Zaragoza, Spain; 4Research Group Sector III Healthcare (GIIS081), Institute of Research of Aragón, 50009 Zaragoza, Spain; 5Saint Joseph’s Catholic Hospital, Monrovia 1000, Liberia; manking4unity53@gmail.com; 6African Women’s Research Observatory, 08009 Barcelona, Spain; gmartinezgabas@gmail.com

**Keywords:** female genital mutilation, female genital cutting, training evaluation, nurses, midwives, obstetric complications, Liberia, Sub-Saharan Africa

## Abstract

Although female genital mutilation/cutting (FGM/C) is a prevalent practice in Liberia, healthcare workers lack the capacity to provide adequate care for FGM/C survivors. Therefore, Liberian nurses, physician assistants, midwives and trained traditional midwives were trained in sexual, obstetric and psychosocial care for FGM/C survivors in 2019. Through questionnaires, we assessed knowledge acquisition, trainee attitudes towards FGM/C care and acceptability to implement WHO-endorsed recommendations. The questionnaires were analyzed using descriptive statistics for quantitative data and an inductive approach for qualitative data. A total of 99 female and 34 male trainees participated. Most trainees perceived FGM/C as harmful to women’s health, as a violation of women’s rights and showed a willingness to change their clinical practice. While 82.8% (*n* = 74/90) perceived their role in advocating against FGM/C, 10.0% (*n* = 9/90) felt that they should train traditional circumcisers to practice FGM/C safely. The pre-training FGM/C knowledge test demonstrated higher scores among physician assistants (13.86 ± 3.02 points) than among nurses (12.11 ± 3.12 points) and midwives (11.75 ± 2.27 points). After the training, the mean test score increased by 1.69 points, from 12.18 (±2.91) points to 13.87 (±2.65) points. The trainings successfully increased theoretical knowledge of FGM/C-caused health effects and healthcare workers’ demonstrated willingness to implement evidence-based guidelines when providing care to FMG/C survivors.

## 1. Introduction

Enhanced and innovative training provisions on obstetric and psychosocial care for female genital mutilation/cutting (FGM/C) survivors is crucial to building healthcare workers’ capacities in FGM/C-affected communities [1]. Evidence demonstrates that FGM/C leads to obstetric complications, such as perineal tears and prolonged labour, and may impede healthcare workers’ assessment and treatment, such as intrapartum vaginal examination or catheterization [2,3,4]. Moreover, FGM/C may cause trauma and have a long-lasting negative psychological impact on women [5,6,7].

Building the healthcare workforce’s capacity to mitigate the impact of FGM/C is not without challenges. In countries such as Egypt, Kenya or Nigeria, where a growing trend towards medicalization of FGM/C has been denounced, some healthcare workers may not accept training from stakeholders advocating for FGM/C abolition [8]. In Guinea, Liberia or Sierra Leone, FGM/C is a requisite for girls to join a secret society, making FGM/C a practice that nobody dares to unveil. The secrecy around FGM/C hampers advocacy activities and education on its health effects [9]. Further, there is a lack of legal, social, and institutional support to tackle training needs and to ensure the translation of knowledge into clinical practice [10].

In Liberia specifically, the secret Sande society demands girls to undergo FGM/C before joining the society as members [11,12]. As the Sande society is powerful in Liberia and influences the socio-religious life greatly, people fear repercussions for openly talking about FMG/C [13]. In the same context, during the discussion of a new domestic violence law in 2016 in the parliament, a clause abolishing FGM/C was removed [14]. A one-year Executive Order (No. 92) on domestic violence was issued in 2018 by the former president, Ellen Johnson Sirleaf [15]. However, the Executive Order was not extended-legalizing FGM/C again [16]. Despite a decrease in the prevalence of FGM/C in Liberia, the average is at 38.2% in women aged 15–49, with the highest prevalence in the North-Western and North-Central regions of 68.3% and 54.2%, respectively [17].

To improve healthcare provision to survivors of FGM/C in Liberia, training tailored to the expectations, needs, capacities, and social and cultural values of the Liberian health workforce must be implemented [1,18]. In ensuring the effectiveness of FGM/C training provision in Liberia, a study of acronym PerTradFGMo was carried out in 2017, which assessed healthcare workers’ experiences with obstetric care provision to FGM/C survivors in Liberia [9]. PerTradFGMo findings informed the design of a training programme titled “Integrated FGM/C Antenatal and Psychosocial Care Workshop” (hereafter workshop) that was conducted in 2019. Alongside the implementation of the workshops, an interim formal evaluation was conducted to assess trainees’ attitudes towards their role as healthcare providers in FGM/C care; acceptability to apply evidence-based recommendations to cater for the obstetric and psychosocial needs of FGM/C survivors and trainees’ knowledge acquisition. This paper reports on the findings of this training evaluation with the aim of discussing practical recommendations and implications for future culturally congruent FGM/C training.

## 2. Materials and Methods

The workshops were a component of the project YOUCANTRY!, a cooperation for healthcare capacity development action led by the Catalonian association, the African Women’s Research Observatory. They were implemented in Liberia from February to September 2019. The evaluation of the workshops was framed as a descriptive, longitudinal, non-experimental study with an ethnographic approach to data collection and analysis in an ex-ante and ex-post manner [19].

### 2.1. Sites and Population

The workshops were conducted in the counties of Montserrado, Nimba, and Lofa, with an FGM/C prevalence in women aged 15–49 of 25.2%, 41.7% and 68.2%, respectively [17]. Obstetric care providers (i.e., physician assistants, registered nurses, certified midwives, and trained traditional midwives (TTM)) working in these counties were invited to the workshops. Local female nurses and a male physician assistant purposely selected the workshops’ trainees.

Based on the time availability of approached trainees, two workshops were organised targeting female TTM only (Lofa-female-TTM, Nimba-female-TTM), two aimed at single-sex and mixed-profession (Lofa-female-mixed, Montserrado-male-mixed), and two at mixed-sex and mixed-profession (Montserrado-mixed-mixed, Nimba-mixed-mixed). All trainees attending the six workshops form the study population.

### 2.2. Training Syllabus

The two-day workshops were led by Liberian female (*n* = 3) and male (*n* = 1) facilitators under the guidance of a female lead trainer from Germany. English was the main language used, except for the workshops targeting TTM, which were conducted in the counties’ autochthonous languages.

The syllabus included education on sexual, obstetric, and psychosocial care for FGM/C survivors. Theoretical content was based on the World Health Organization handbook “Care of girls and women living with female genital mutilation” [20]. Exercises and group discussions accompanied theory sessions to improve knowledge acquisition. A mobile phone application to support the translation of the handbook’s contents into clinical practice was designed and introduced to the trainees during the workshops.

### 2.3. Data Collection and Analysis

At the start of each workshop, trainees received a verbal explanation about the evaluation methodology to assess the effectiveness of the workshop’s approach to improving local capacities to care for FGM/C survivors. Trainees were asked to verbally confirm their willingness to collaborate in the evaluation of the training program.

The following data collection tools were administered only to the trainees who agreed to collaborate in the training evaluation:

Attitudes Questionnaire. This questionnaire assessed healthcare workers’ attitudes toward cultural values around FGM/C, its effect on women’s health, and healthcare workers’ roles and attitudes toward medicalization. It consisted of four single and one multiple choice question and was administered either at the end of the first or the beginning of the second workshop day. Attitudes questionnaires were digitized into Microsoft Excel^®^ (RRID: SCR_016137, Microsoft Corporation, Redmond, WA, USA). Descriptive analysis for aggregated attitudes data was run in Stata^®^ v. 16.0 (RRID: SCR_012763, StataCorp LLC, College Station, TX, USA).Satisfaction Questionnaire. This questionnaire aimed to understand the adequacy and cultural competency of the training materials, the workshop facilitators, and the pedagogic approach. In 13 questions, it asked trainees to comment on their perception of the workshop’s effectiveness for knowledge acquisition and translation into practice. Five questions used a Likert scale (1 = very unsatisfied, 5 = very satisfied), five were open-ended, and three were single choice. They were administered at the end of the workshop. Satisfaction questionnaires’ quantitative and qualitative data were digitised into Microsoft Excel^®^ (RRID: SCR_016137) and Word^®^ (Microsoft Corporation, Redmond, WA, USA), respectively. Descriptive analyses of aggregated quantitative data were run in Stata^®^ v. 16.0 (RRID: SCR_012763), while qualitative data was subjected to inductive analyses for core concepts of interest in Microsoft Word^®^.Pre- and post-training knowledge questionnaire. Pre- and post-training questionnaires were identical. These were the only non-anonymous data collection tools administered to the trainees. The pre-training questionnaire was administered at the beginning of the first workshop day and the post-training questionnaire at the end of the workshop. In the first of its three sections, trainees’ self-perceived knowledge, confidence and skills to care for FGM/C survivors were assessed, using 10 rating items (1 = strongly disagree, 5 = strongly agree). The following sections assessed trainees’ general knowledge of FGM/C (eight single-choice questions), genital assessment (three single-choice questions), and FGM/C-related obstetric complications (nine single-choice questions). Each correct answer (i.e., excluding the first section) counted one point towards the final test score (max. 20 points). Knowledge questionnaires were digitized into Microsoft Excel^®^ (RRID:SCR_016137). Descriptive analysis for aggregated data was run in Stata^®^ v. 16.0 (RRID:SCR_012763). Only the questionnaires of registered nurses, certified midwives and physician assistants were included in the analysis (*n* = 75). After excluding questionnaires of trainees who did not complete both questionnaires (*n* = 4), the analysis was based on a total of 71 sets of pre- and post-training questionnaires. Invalid responses (e.g., various responses marked) were treated as no response. The pre- and post-training knowledge questionnaires were matched, and bivariate paired t-tests were conducted to assess the effects of gender, age, occupation, work setting and workshop setting on changes in test score differences, using a significance level of *p* < 0.05. 95% confidence intervals were calculated.

Due to the high number of illiterate trainees in the Lofa-female-TTM and Nimba-female-TTM workshops, both anonymous questionnaires (i.e., attitudes and satisfaction questionnaire) of these workshops were not included in the analysis. However, responses of illiterate trainees from other workshops to the anonymous questionnaires were included in the analysis.

### 2.4. Triangulation

The lead trainer practiced reflexivity, using an observation guide and a field diary, throughout the conduct of all workshops. Reflexivity helped to assess the effect of trainers’ interaction with the trainees on their narratives and engagement in the training. Field notes were used for quality control, to detect social desirability bias, to monitor for social harm, and for triangulation.

### 2.5. Ethics

Ethics approval was granted in January 2019 by the University of Liberia-Pacific Institute for Research Evaluation Institutional Review Board (Ref. #19-01-148).

To avoid social harm, trainees were not encouraged to discuss their personal opinions on the Sande society. All questionnaires relating to trainees’ personal beliefs (attitudes and satisfaction questionnaires) were anonymous.

Strict data processing measures were implemented. Personal identifiers were neither transcribed into any document, nor shared with third parties nor reported in any output. To prevent identification of trainees, only aggregated socio-demographic data are presented in this article. Handwritten diary pages were destroyed once transcribed, and questionnaires are kept in a locked cabinet in the premises of the principal investigator’s institution.

## 3. Results

### 3.1. Sociodemographic Characteristics

A total of 133 trainees participated in the workshops (Table 1). Most trainees were female (71.2%), with a mean age of 43.0 years, as compared to 33.5 years for males. Almost all males (91.2%) attended the Montserrado-mixed-mixed workshop, while 25 (25.3%) females attended the workshop held in Montserrado, 33 (33.3%) in Nimba, and 41 (41.4%) in Lofa. Among the females, 43 TTM (43.4%), 27 registered nurses (27.3%), and 20 certified midwives (20.2%) attended. Most male trainees were registered nurses (*n* = 21, 61.8%) and physician assistants (*n* = 7, 20.6%). Nearly half of the males (*n* = 16, 47.1%) worked in a hospital, while about two thirds of the females worked in primary healthcare (*n* = 64, 64.6%). Twelve of the 39 TTM (30.8%) with known literacy status were literate (*n* = 4 unknown literacy status).

### 3.2. Attitudes

Ninety trainees who attended the Lofa-female-mixed, Montserrado-mixed-mixed, Montserrado-male-mixed and Nimba-mixed-mixed workshops filled out the anonymous attitudes questionnaire (Table 2). Overall, 97.8% of the questionnaire respondents perceived FGM/C as harmful to women’s health, and 93.3% as a human rights violation. In the Lofa-female-mixed workshop, 14.3% (*n* = 3/21) of respondents did not perceive FGM/C as a women’s rights violation as compared to 0.0% in Nimba and 5.5% (*n* = 3/55) in Montserrado. Most men in the Montserrado-male-mixed workshop (*n* = 28/29, 96.6%) agreed that FGM/C negatively affected women’s health and viewed it as a violation of human rights.

The majority of respondents (76.7%) expressed that FGM/C should be stopped. Nevertheless, 15.6% (*n* = 14/90) thought that FGM/C was indispensable to join the Sande society. Two out of 21 (9.5%) of the Lofa-female-mixed respondents opined that FGM/C should continue to be a requisite to enter the Sande society as opposed to 0.0% of trainees in Montserrado and Nimba.

Most healthcare workers (*n* = 74/90, 82.8%) acknowledged their role in advocating against FGM/C. A minority (*n* = 9, 10.0%) indicated that they should train “zoes” (Sande traditional circumcisers, also referred to as “showeis” in the literature) on how to practice FGM/C safely. Of these nine, five were trainees from Montserrado-male-mixed.

Likewise, eleven trainees (12.2%) agreed that healthcare workers should be invited to practice FGM/C in the bush, and five (5.6%) trainees from Montserrado thought that FGM/C should be practiced by healthcare workers in the clinics. In comparing the male-only and female-only workshops, male trainees (17.2%) were more likely than female trainees (14.3%) to support that healthcare workers should be invited by the “zoes” to practice FGM/C in the Sande initiation camps. Similarly, 10.3% of male trainees (females: 0.0%) thought that they should be the ones practicing FGM/C in the clinics and hospitals.

After the training, one trainee stated in the satisfaction questionnaire that FGM/C was needed to “*educate our women*” (Montserrado-male-mixed), while most trainees acknowledged the harmful health-effects of FGM/C. Trainees emphasised the importance of patient education about FGM/C and motivating patients to share the knowledge with their communities. Most trainees seemed confident that, through health education, people might be convinced to stop supporting FGM/C. However, one trainee stated that “*no matter what you do or say to them cannot change anything*” (Montserrado-mixed-mixed).

### 3.3. Satisfaction

Ninety-one trainees attending the Lofa-female-mixed, Montserrado-mixed-mixed, Montserrado-male-mixed and Nimba-mixed-mixed workshops anonymously responded to the satisfaction questionnaire (Table 3). Overall, trainees were satisfied with the course (4.51 ± 0.57/5.00), the quality of course contents (4.37 ± 0.71/5.00), and the course facilitators (4.76 ± 0.46/5.00). Trainees in Lofa-female-mixed scored the general experience slightly lower (4.30 ± 0.73/5.00), and trainees of Montserrado-mixed-mixed slightly higher (4.71 ± 0.46/5.00). Concerning the course quality, trainees in Nimba-mixed-mixed scored slightly lower (4.17 ± 0.39/5.00) as compared to the other trainees, albeit rating only 4 or 5 as opposed to the other workshops where the range was broader.

Almost all trainees (*n* = 73) would recommend the course to others. Trainees highly rated the utility of knowledge and information gained (4.48 ± 0.59/5.00). The average rating of the workshops‘ relevance for FGM/C capacity building was 4.29 ± 0.85/5.00. Trainees of Nimba-mixed-mixed rated the workshops as slightly more relevant (4.46 ± 0.52/5.00). Most trainees (*n* = 79) manifested a willingness to implement changes in antenatal and psychosocial care provision for FGM/C survivors.

In the satisfaction questionnaire, one trainee criticised the “*exposure*” of Liberian culture. Course facilitators were perceived as well-prepared to address the needs of all cadres of healthcare professions. Trainees appreciated the interactive structure of the course, especially the group works, where they had the possibility to practice their newly acquired knowledge in the form of a drama.

Nearly all trainees would recommend the workshops to others. As one trainee from Nimba expressed: “*I will recommend this course to my friend so that we all are going to talk against FGM/C in Liberia*”. In both single-gender trainings, trainees expressed the importance to “*involve the opposite sex*”. In all workshops, it was recommended to extend the training to other facilities to improve their colleagues’ skills to look after FGM/C survivors. One trainee from Nimba (gender unknown) suggested offering the workshops to “*all health worker so that they may be able to understand the danger of FGM and work properly with girls and women affected*”. One female trainee from Lofa felt empowered to “*mentor those facilities staff that was not part of this training in the district*”. Trainees recommended teaching the workshops “*in all medical institutions*” and engaging “*traditional leaders, pastors, teachers and high school student*”.

### 3.4. Knowledge and Skills Acquisition

Seventy-one matched pre- and post-training knowledge questionnaires were included in the analysis.

#### 3.4.1. Self-Perceived Confidence, Knowledge, and Skills in FGM/C Survivor Care

Pre-training self-perceived appraisal of all items concerning confidence, knowledge and skills in FMG/C survivor care was similar across workshops and ranged between 3.18 (knowledge on treatment of FGM/C-related complications during pregnancy, max. 5 points) and 3.94 (confidence in clinical history taking, max. 5 points).

After participation in the workshops, trainees reported to be more confident in clinical history taking (Δ = 0.55 ± 1.12, 95% CI: 0.28–0.83), performing a genital examination (Δ = 0.64 ± 1.58, 95% CI: 0.26–1.02), and providing obstetric care (Δ = 0.64 ± 1.47, 95% CI: 0.28–0.99) (Table 4). Trainees self-perceived as more knowledgeable on which questions to ask to identify FGM/C-attributable complications (Δ = 0.94 ± 1.49, 95% CI: 0.58–1.30), on how to identify complications during genital examination (Δ = 0.93 ± 1.44, 95% CI: 0.58–1.27), and on how to treat them (Δ = 0.85 ± 1.30, 95% CI: 0.53–1.17). They reported that they self-perceived as more skilled to perform a genital examination (Δ = 0.85 ± 1.43, 95% CI: 0.51–1.20) and to distinguish the different FGM/C types (Δ = 0.87 ± 1.40, 95% CI: 0.53–1.21).

After the workshop, trainees from the Montserrado-mixed-mixed workshop felt less enabled (Δ = −0.20 ± 1.26, 95% CI: −0.90–0.50, not statistically significant) to identify adverse conditions in pregnancy and childbirth, while trainees of the Nimba-mixed-mixed workshop felt less enabled (Δ = −0.11 ± 1.05, 95% CI: −0.92–0.70, not statistically significant) after the workshop to perform a genital examination.

Self-perceived pre-workshop confidence across all items was higher for male trainees, whereas self-perceived pre-workshop knowledge for all items was higher for women. Self-perceived pre-workshop skills were similar between both gender groups. After the workshops, male trainees showed a higher increase than female trainees in knowledge- and skills-items, whereas female trainees showed a higher increase than male trainees in their confidence in providing obstetric care, developing a birth plan, and taking a clinical history.

Physician assistants reported a higher self-perceived pre-workshop confidence in providing obstetric care and performing a genital examination than registered nurses (who, in turn, showed a higher confidence than certified midwives). Registered nurses showed more pre-workshop confidence in clinical history taking and more pre-workshop knowledge of the management of FGM/C complications than physician assistants and certified midwives. Certified midwives reported to be more knowledgeable in identifying FGM/C complications through questions and genital examinations than registered nurses and physician assistants in the pre-workshop questionnaire. All occupation groups reported a similar ability to perform a genital examination in the pre-workshop questionnaire. Registered nurses reported in the pre-workshop questionnaire the highest score on skills to identify and record removed genitalia parts, while certified midwives felt more enabled to look for conditions that may lead to problems during pregnancy and childbirth than the other two groups. 

#### 3.4.2. Single-Choice Knowledge Questions

After the training, the mean test score on knowledge items increased from 12.18 points (±2.91) points to 13.87 points (±2.65) (95% CI: 1.13–2.25) (Table 5). Male trainees achieved a higher pre-test score (13.14 ± 2.76) than female trainees (11.56 ± 2.87), whose post-test result increased on average 1.84 points as compared to 1.46 points in men. While the mean increase in points across the ages was similar, participants older than 45 years scored on average two points lower than their fellow trainees of younger age groups. The pre-training test score was higher among physician assistants (13.86 ± 3.02) than among registered nurses (12.11 ± 3.12) and certified midwives (11.75 ± 2.27). In contrast, certified midwives improved in their post-training evaluation the most, by 2.45 points (95% CI: 1.50–3.40), registered nurses by 1.52 points (95% CI: 0.76–2.29) and physician assistants the least, by 0.57 points (95% CI −0.93–2.07, not statistically significant). Hospital-based trainees had, on average, a 0.75 point lower pre-test than primary healthcare-based trainees. However, hospital-based trainees increased their score by an average of 2.38 points (95% CI: 1.64–3.11) to a score of 14.16 points (±1.99), surpassing primary healthcare-based trainees by 0.53 points. The trainees of the Nimba-mixed-mixed workshop had the highest pre-test score across all workshops but did not show any improvement in their post-test score. The Monrovia-mixed-mixed trainees reached the lowest pre-test (10.56 ± 3.03 points) and post-test score (12.17 ± 3.11 points) of all workshops.

Some trainees expressed in the satisfaction questionnaire that they valued the knowledge gained about the health effects of FGM/C and their management. They stated that there was a positive influence of the knowledge acquired on the entire interaction cascade with FGM/C survivors. According to some, the training enabled them to “*enhance*” their history-taking skills, to conduct an exhaustive ANC assessment with screening for FGM/C, and to anticipate its complications. Trainees felt that they had improved their capacities to treat the physical and psychological effects of FGM/C, and they manifested that they were aware of referral needs for further management to decrease maternal mortality. Some male trainees pointed out that the course enabled them to have a better rapport and improved communication with the FGM/C survivors. Trainees emphasised that the workshops showed them how to provide health education to FGM/C survivors about “*how to cope*” with their situation.

## 4. Discussion

The workshops successfully increased self-perceived knowledge and skills in detection and management of FGM/C-related complications, as well as theoretical knowledge on FGM/C and its health effects. Healthcare providers adopted the WHO-endorsed recommendations, and manifested a willingness to apply them during their encounters with FGM/C survivors. Although the traditional practice of FGM/C is pervasive in Liberia, this study suggests that most healthcare professionals are aware of the need to abstain from it.

Our findings reflect that FGM/C survivor care is not adequately addressed in institutional healthcare professional training. Trainees recommended including education on FMG/C management in nursing and midwifery curricula and extending mixed-gender workshops to other cadres of healthcare providers. Literature affirms that innovative education as our workshops increase FGM/C-knowledge [1] and integration into professional training curricula is necessary [21]. While we did not evaluate behavioural changes, knowledge and perceived skills improved. Our data contributes to a clearer understanding of healthcare workers’ attitudes towards FGM/C, which influences their care provision to FGM/C survivors [4,22]. A clear understanding of attitudes can inform further training designs and might prevent stigmatisation or double discrimination against FGM/C survivors [23,24], enabling healthcare providers to create an appropriate rapport with the patient.

Our results showed that while the vast majority of healthcare workers perceived FGM/C as a human rights violation, one out of ten healthcare providers supported FGM/C medicalisation. This is to be seen critically from an ethical perspective [8] and might lead to disagreements between healthcare workers within healthcare facilities if no guidelines are present [25]. According to Kimani (2020), some healthcare providers suggested that medicalization inhibits the abandonment of the practice and instead “modernised” it, boosting healthcare providers’ income, and incentivizing parents to cut their daughters [26]. Scholars have argued that medicalisation is a tacit approval for FGM/C, which creates the impression that FGM/C can be performed safely, and that it is condoned by respected health care professionals and institutions, thus reducing families’ motivation to abandon the practice [27,28].

In abolishing FGM/C, the role of healthcare workers might not only be as multiplicators within their community but also as promotors for political decisions [29]. However, in the Liberian context, this may prove difficult. Healthcare professionals are usually not the most senior and may come from the capital region, thus they might not be the most respected of the community. The Sande society might impede nurses and midwives from advocating at the community level and taking part in the decision-making process. Therefore, even if the training proved to be effective in shaping healthcare providers’ attitudes, it is not a given that a social transformation will take place.

If more programmes involved the knowledge, skills and attitudes approach that our workshops defended, a more nuanced picture of what works and for whom could be obtained. This could lead to quicker progress towards implementing the most effective strategies for change. Studies have shown that specialised training for healthcare providers improves healthcare workers’ knowledge of FGM/C [30] and suggest that successful abandonment of FGM/C requires that healthcare professionals be fully equipped to address prevention and management of the practice [31]. In line with other studies, this study showed that advocacy activities should be extended to other stakeholders, such as traditional and church leaders, to abstain from FGM/C in the community and to political stakeholders to delegalize FGM/C [29,32].

### Limitations

Methodological choices were constrained by the high anticipated illiteracy rate in the TTM sub-group. Despite the satisfaction and attitudes questionnaires being voluntary, the responses of trainees in the questionnaires might be limited by social desirability bias. A few trainees had difficulties with English language and needed help to fill them out, which may have influenced their responses. Finally, caution is warranted not to generalise the results of the knowledge questionnaire to other settings. It must be noted that, as there was no control group in this evaluation, regression to the mean could not be measured [33].

## 5. Conclusions

This study shows that the workshops were successful in increasing healthcare workers’ knowledge and self-perceived skills in providing FGM/C care. The identified attitudes towards FGM/C and healthcare workers’ recommendations might inform future training designs and has the potential to accelerate the abandonment of FGM/C. Further research is needed to evaluate acquired knowledge and skills transfer from a theoretical setting to clinical practice.

## Figures and Tables

**Table 1 ijerph-19-09209-t001:** Socio-demographics of trainee participants.

		Lofa-Female-Mixed (Aug’19)	Lofa-Female-TTM (Aug’19) ^†^	Montserrado-Mixed-Mixed (Feb’19)		Montserrado-Male-Mixed (Sep’19)	Nimba-Mixed-Mixed (Feb’19)		Nimba-Female-TTM (Feb’19) ^†^	Total
		F, *n* = 21	F, *n* = 20	F, *n* = 25	M, *n* = 2	M, *n* = 29	F, *n* = 11	M, *n* = 3	F, *n* = 22	*n* = 133
Age	<35	7	5	8	2	15	0	1	6	44 (33.1%)
	35–44	10	2	8	0	11	6	2	6	45 (33.8%)
	≥45	4	13	9	0	3	3	0	9	41 (30.8%)
	NA	0	0	0	0	0	2	0	1	3 (2.3%)
Occupation	Certified Midwife	14	0	6	0	0	0	0	0	20 (15.0%)
	Physician Assistant	0	0	0	0	7	0	0	0	7 (5.3%)
	Registered Nurse	7	0	14	2	16	6	3	0	48 (36.1%)
	TTM	0	14	4	0	0	3	0	22	43 (32.3%)
	Other ^‡^	0	6	0	0	6	2	0	0	14 (10.5%)
	NA	0	0	1	0	0	0	0	0	1 (0.8%)
Work environment	Hospital	14	0	0	0	16	2	0	0	32 (24.1%)
	Primary Care	6	3	25	2	8	9	3	21	77 (57.9%)
	Community	0	17	0	0	0	0	0	0	17 (12.8%)
	NA	1	0	0	0	5	0	0	1	7 (5.3%)
Literacy	Literate	21	4	5	0	29	7	3	11	80 (60.2%)
	Illiterate	0	16	0	0	0	3	0	11	30 (22.6%)
	NA	0	0	20	2	0	1	0	0	23 (17.3%)

^†^ Conducted in local languages, ^‡^ such as Community Health Volunteer, Nurse Assistant, Health Science Student. *n* = Number of participants, NA = No data available, F = Female, M = Male, TTM = Trained Traditional Midwife.

**Table 2 ijerph-19-09209-t002:** Healthcare workers’ attitudes towards Female Genital Mutilation/Cutting.

		Lofa-Female-Mixed *n* = 21		Montserrado-Mixed-Mixed *n* = 26		Montserrado-Male-Mixed *n* = 29		Nimba-Mixed-Mixed *n* = 14		Total *n* = 90	
		*n*	%	*n*	%	*n*	%	*n*	%	*n*	%
FGM/C is a violation of human rights.	No	3	14.3	2	7.7	1	3.4	0	0.0	6	6.7
	Yes	18	85.7	24	92.3	28	96.6	14	100.0	84	93.3
FGM/C is harmful to women’s health.	No	0	0.0	1	3.8	1	3.4	0	0.0	2	2.2
	Yes	21	100.0	25	96.2	28	96.6	14	100.0	88	97.8
FGM/C needs to be continued as a requisite for girls and women to become a member of the Sande society.		2	9.5	0	0.0	0	0.0	0	0.0	2	2.2
FGM/C needs to be replaced by a non-harmful symbolic practice (e.g., pricking).		1	4.8	5	19.2	9	31.0	4	28.6	19	21.1
FGM/C needs to be stopped and not replaced by any other type of practice.		18	85.7	21	80.8	20	69.0	10	71.4	69	76.7
It should be indispensable that girls and women undergo FGM/C to join the Sande society.	No	18	85.7	22	84.6	22	75.9	14	100.0	76	84.4
	Yes	3	14.3	4	15.4	7	24.1	0	0.0	14	15.6
Healthcare professionals must train showeis to practice FMG/C safely in the bush.	No	19	90.5	24	92.3	24	82.8	14	100.0	81	90.0
	Yes	2	9.5	2	7.7	5	17.2	0	0.0	9	10.0
Healthcare professionals must be invited by showeis to practice FGM/C themselves in the bush.	No	18	85.7	23	88.5	24	82.8	14	100.0	79	87.8
	Yes	3	14.3	3	11.5	5	17.2	0	0.0	11	12.2
Healthcare professionals must be the ones practicing FGM/C in clinics and hospitals.	No	21	100.0	24	92.3	26	89.7	14	100.0	85	94.4
	Yes	0	0.0	2	7.7	3	10.3	0	0.0	5	5.6
Healthcare professionals must campaign and/or advocate against FGM/C.	No	4	19.0	7	26.9	5	17.2	0	0.0	16	17.8
	Yes	17	81.0	19	73.1	24	82.8	14	100.0	74	82.8

FGM/C = Female Genital Mutilation/Cutting, *n* = number of participants, showeis = Sande traditional circumcisers.

**Table 3 ijerph-19-09209-t003:** Satisfaction of trainees regarding the workshops.

	Lofa-Female-Mixed *n* = 21			Montserrado-Mixed-Mixed *n* = 27			Montserrado-Male-Mixed *n* = 29			Nimba-Mixed-Mixed *n* = 14			Total *n* = 91		
	Mean	SD	NA	Mean	SD	NA	Mean	SD	NA	Mean	SD	NA	Mean	SD	NA
Overall	4.30	0.73	1	4.71	0.46	6	4.52	0.51	0	4.50	0.52	2	4.51	0.57	9
Relevance	4.20	0.95	1	4.32	0.69	2	4.25	1.04	1	4.46	0.52	1	4.29	0.85	5
Quality	4.42	0.77	2	4.39	0.89	4	4.39	0.63	1	4.17	0.39	2	4.37	0.71	9
Utility	4.38	0.50	0	4.48	0.79	4	4.50	0.51	1	4.57	0.51	0	4.48	0.59	5
Facilitators	4.81	0.40	0	4.79	0.51	3	4.68	0.48	1	4.79	0.43	0	4.76	0.46	4

*n* = Number of participants, NA = No data available, SD = Standard deviation.

**Table 4 ijerph-19-09209-t004:** Changes in trainees’ self-perceived confidence, knowledge, and skills on care provision to FGM/C survivor post- vs. pre-training.

		Lofa-Female-Mixed *n* = 21			Montserrado-Mixed-Mixed *n* = 18			Montserrado-Male-Mixed *n* = 23			Nimba-Mixed-Mixed *n* = 9			Total *n* = 71			
		Mean Difference	SD	NA	Mean Difference	SD	NA	Mean Difference	SD	NA	Mean Difference	SD	NA	Mean Difference	SD	NA	*p*-Value
Confidence	Providing obstetric care	0.86	1.88	0	0.19	1.28	2	0.70	1.26	0	0.78	1.30	0	0.64	1.47	2	0.00
	Developing a birth plan	1.52	1.69	0	0.47	1.60	3	0.83	1.19	0	0.44	1.13	0	0.91	1.48	3	0.00
	Taking a clinical history during pregnancy	0.95	1.43	2	0.31	1.01	2	0.48	0.79	0	0.33	1.22	0	0.55	1.12	4	0.00
	Performing a genital examination during pregnancy	0.24	1.84	0	1.31	1.62	2	0.74	0.96	0	0.11	1.90	0	0.64	1.58	2	0.00
Knowledge	Identifying obstetric FMG/C complications during history taking	0.95	1.53	0	0.75	1.44	2	1.17	1.59	0	0.67	1.41	0	0.94	1.49	2	0.00
	Identifying obstetric FGM/C complications when performing a genital examination	0.71	1.06	0	0.80	2.11	3	1.30	1.29	0	0.67	1.22	0	0.93	1.44	3	0.00
	Treating obstetric FGM/C complications	1.00	1.38	1	0.21	1.12	4	1.09	1.38	0	0.89	1.05	0	0.85	1.30	5	0.00
Skills	Performing a genital examination	1.15	1.42	1	0.56	1.50	2	1.17	1.37	0	−0.11	1.05	0	0.85	1.43	3	0.00
	Distinguishing FGM/C types by identifying and recording removed genitalia	1.48	1.36	0	0.20	1.32	3	0.96	1.30	0	0.33	1.41	0	0.87	1.40	3	0.00
	Identifying conditions likely to cause obstetric problems	0.86	1.35	0	−0.20	1.26	3	1.09	1.12	0	0.56	0.53	0	0.66	1.25	3	0.00

FGM/C = Female Genital Mutilation/Cutting, Mean = Mean change between post- and pre-test score, *n* = number of participants, NA = No data available, SD = Standard deviation.

**Table 5 ijerph-19-09209-t005:** Mean and mean difference of pre- and post-training results stratified by socio-demographics.

		Total		Pre-Training Test Result		Post-Training Test Result		Difference between Post- and Pre-Training Test Result		Test Parameter
		*n*	%	Mean	SD	Mean	SD	Mean Difference	95% CI	*p*-Value
**Overall**		71	100	12.18	2.91	13.87	2.65	1.69	1.13–2.25	0.00
Gender *n* = 71	Female	43	60.6	11.56	2.87	13.40	2.57	1.84	1.09–2.58	0.00
	Male	28	39.4	13.14	2.76	14.61	2.64	1.46	0.58–2.35	0.00
Age *n* = 70	<35	27	38.6	12.52	3.14	14.04	3.20	1.52	0.55–2.50	0.00
	35–44	31	44.3	12.39	2.85	14.19	1.99	1.81	0.98–2.62	0.00
	≥45	12	17.1	10.58	2.02	12.42	2.47	1.83	0.17–3.50	0.03
Occupation *n* = 71	Physician Assistant	7	9.9	13.86	3.02	14.43	2.51	0.57	−0.93–2.07	0.39 *
	Registered Nurse	44	62.0	12.11	3.12	13.64	3.02	1.52	0.76–2.29	0.00
	Certified Midwife	20	28.2	11.75	2.27	14.20	1.70	2.45	1.50–3.40	0.00
Setting *n* = 70	Primary Healthcare	38	54.3	12.53	3.29	13.63	3.14	1.11	0.28–1.93	0.01
	Hospital	32	45.7	11.78	2.45	14.16	1.99	2.38	1.64–3.11	0.00
Workshop *n* = 71	Nimba_mixed_mixed	9	12.7	14.67	2.74	14.67	3.00	0.00	−1.49–1.49	1.00 *
	Monrovia_ mixed_mixed	18	25.4	10.56	3.03	12.17	3.11	1.61	0.26–2.97	0.02
	Lofa_female_mixed	21	29.6	11.67	2.20	14.14	2.08	2.48	1.35–3.61	0.00
	Monrovia_male_mixed	23	32.4	12.96	2.69	14.65	2.08	1.70	0.92–2.47	0.00

*p*-values marked with an asterisk (*) are not statistically significant. CI = Confidence Interval, *n* = Number of participants, SD = Standard deviation.

## Data Availability

The anonymized quantitative data presented in this study are available in this public repository: https://doi.org/10.17632/97r45by53y.1. The qualitative data that supported this study cannot be made available as per research ethics explained to participants during the informed consent process.

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
