# Peer review of "Management of Female Genital Mutilation/Cutting-Related Obstetric Complications: A Training Evaluation"

_ijerph, 2022, doi:10.3390/ijerph19159209_

Round 1
Reviewer 1 Report
VERY WELL WRITTEN ,NO REMARKS
Author Response
Manuscript ID: ijerph-1805033, Round 1
Responses to reviewer 1
VERY WELL WRITTEN ,NO REMARKS
Reply: We thank the reviewer for appreciating our work.
On behalf of all authors,
Kim Nordmann
Reviewer 2 Report
This paper increases the readers ' awareness of the situation of female genital mutilation in some countries and also the effort to educate local health care workers on this topic.
I have the following minor comments
1. From Table 2, it appeared that the improvements in scores for some areas were only slight. The mean score increase in many areas were less than 1 point. Would the authors comment if these improvements were statistically or clinically significant ?
2. At baseline, 97.8% of the participants already perceived FGM as harmful. It appears that the course should be more targeted to people who still perceived FGM as a necessity. ( ie it seemed to be " preaching to the converted" )
3. In the results section, the scores were reported according to the different ages/ professions/ areas etc. Would the authors discuss what is their significance or implications and what conclusion can the authors derive from analysing the scores according to these groups? eg Can we obtain any information on, for example, is it better to conduct the training in primary health or hospital setting etc
4. The discussion is mainly on the attitude on FGM instead of the effect on knowledge improvement from the course. It is a bit confusing as to what the main purpose of the course is . Is it to improve the health care worker's competence in caring for FGM survivors or is it to advocate against FGM ?
Author Response
Manuscript ID: ijerph-1805033, Round 1
Responses to reviewer 2
Reply: We thank you for your extensive and helpful review. We have addressed your observations and suggestions in a point-by-point manner below and highlighted them in the manuscript using track changes.
Comments and Suggestions for Authors
This paper increases the readers ' awareness of the situation of female genital mutilation in some countries and also the effort to educate local health care workers on this topic.
I have the following minor comments
- From Table 2, it appeared that the improvements in scores for some areas were only slight. The mean score increase in many areas were less than 1 point. Would the authors comment if these improvements were statistically or clinically significant ?
Reply: Thank you for pointing this out. We included in the manuscript 95% confidence intervals and the overall p-values for each line. As can be seen, the overall self-reported improvements of all trainees in the different categories (skills, confidence and knowledge) are statistically significant.
- At baseline, 97.8% of the participants already perceived FGM as harmful. It appears that the course should be more targeted to people who still perceived FGM as a necessity. ( ie it seemed to be " preaching to the converted" )
Reply: Thank you very much for this comment. In Liberia, girls and women undergo FGM/C as an initiation ritual to a secret society. Due to the secrecy, little is taught in paramedical school about the negative health consequences of FGM/C. Therefore, while the vast majority of participants perceived FGM/C as a harmful practice, training is needed to provide knowledge and skills to manage the harmful effects of FGM/C on sexual, obstetric and psychosocial health. Nevertheless, we completely agree with you that trainings on the effects of FGM/C should be extended to other stakeholders, such as traditional and church leaders to abstain from FGM/C in the community and to political stakeholders to delegalize FGM/C, as well as to those healthcare providers that still perceive FGM/C as a necessity.
- In the results section, the scores were reported according to the different ages/ professions/ areas etc. Would the authors discuss what is their significance or implications and what conclusion can the authors derive from analysing the scores according to these groups? eg Can we obtain any information on, for example, is it better to conduct the training in primary health or hospital setting etc
Reply: We appreciate this comment. From the analysis of the pre- and post-training results it becomes apparent that a training i) targeting elderly healthcare providers within the scope of a continuous paramedical education and ii) certified midwives who serve as the main point of contact for obstetric care should be designed to improve care provision to FGM/C survivors.
- The discussion is mainly on the attitude on FGM instead of the effect on knowledge improvement from the course. It is a bit confusing as to what the main purpose of the course is . Is it to improve the health care worker's competence in caring for FGM survivors or is it to advocate against FGM ?
Reply: We thank the reviewer for this comment. Due to political reasons, we decided to design a training to improve the healthcare providers’ competence in managing the harmful effects of FGM/C on sexual, obstetric and psychosocial health. Nevertheless, it is important to understand healthcare providers’ attitudes to design an effective and culturally sensitive training program and to understand healthcare providers’ roles as change agents.
With the changes made we now hope that the manuscript will be suitable for publication in IJERPH.
On behalf of all authors,
Kim Nordmann
Reviewer 3 Report
The study IFAPCW is a component of the project YOUCANTRY, a multinational cooperation led by the Catalonian association African Women's Research Observatory. The studies were implemented in Liberia from February to 76 September 2019. Methodological choices were constrained by the high illiteracy rate. Some trainees had difficulties with English language. It is also warranted not to generalize the 369 results of the knowledge questionnaire to other settings. There was no control group in the evaluation. Even so, there is strong evidence that the IFAPCW were successful in increasing healthcare workers’ 374 knowledge and self-perceived skills in providing FGM/C care.
Author Response
Manuscript ID: ijerph-1805033, Round 1
Responses to reviewer 3
The study IFAPCW is a component of the project YOUCANTRY, a multinational cooperation led by the Catalonian association African Women's Research Observatory. The studies were implemented in Liberia from February to 76 September 2019. Methodological choices were constrained by the high illiteracy rate. Some trainees had difficulties with English language. It is also warranted not to generalize the 369 results of the knowledge questionnaire to other settings. There was no control group in the evaluation. Even so, there is strong evidence that the IFAPCW were successful in increasing healthcare workers’ 374 knowledge and self-perceived skills in providing FGM/C care.
Reply: We thank the reviewer for appreciating our work and pointing out that - albeit methodological constraints - the results indicate that the IFAPCWs were successful in increasing healthcare providers’ knowledge about FGM/C care.
On behalf of all authors,
Kim Nordmann